# In-sensor human gait analysis with machine learning in a wearable microfabricated accelerometer
Guillaume Dion [1], Albert Tessier-Poirier [1], Laurent Chiasson-Poirier[1], Jean-François Morissette [1], Guillaume Brassard[1], Anthony Haman[1], Katia Turcot[2] & Julien Sylvestre [1] ✉

In-sensor computing could become a fundamentally new approach to the deployment of machine learning in small devices that must operate securely with limited energy resources, such as wearable medical devices and devices for the Internet of Things. Progress in this field has been slowed by the difficulty to find appropriate computing devices that operate using physical degrees of freedom that can be coupled directly to degrees of freedom that perform sensing. Here we leverage reservoir computing as a natural framework to do machine learning with the degrees of freedom of a physical system, to show that a micro-electromechanical system can implement computing and the sensing of accelerations by coupling the displacement of suspended microstructures. We present a complete wearable system that can be attached to the foot to identify the gait patterns of human subjects in real-time. The computing efficiency and the power consumption of this in-sensor computing system is then compared to a conventional system with a separate sensor and digital computer. For similar computing capabilities, a much better power efficiency can be expected for the highly-integrated in-sensor computing devices, thus providing a path for the ubiquitous deployment of machine learning in edge computing devices.

The design flexibility of micromechanical devices has enabled a vigorous research effort to develop mechanical computing elements[1], such as memory cells[2] or logic gates[3]. These computing elements are built, for instance, using contact switches or resonators, and they are small, operate at high frequencies, and dissipate a small amount of energy. They might outperform their conventional, electronic counterparts in future computing applications that operate on small power budgets, including battery-powered sensors that perform edge computing in wearable devices or for the Internet of Things[4].

By virtue of their operating in the mechanical domain, mechanical computing devices are especially interesting for their integration with sensors that measure sound, acceleration, strain, shape, adsorbed mass, or other mechanical properties. Ultimately, the mechanical computing device in a sensor could be driven directly by a measured mechanical stimulus to produce without transduction and, in the most efficient manner, a processed output, such as a detection, classification, or control signal. Examples that have been discussed in the literature include resonating structures that are sensitive to sound and can identify certainly spoken commands[5],

metamaterial structures that can identify specific shapes from the positions of a set of contact points[6], as well as various proposals for *in materio* computing[7]. The defining characteristic of these devices is that they can perform complex data processing, including the prediction or the classification of time-varying signals, with minimal or no electronic components that, when they are required, do not contribute substantially to the computing functions of the system. The term *in-sensor computing* has been used to refer to the information processing of physical stimuli by devices that perform both the sensing and the processing of these stimuli in the same physical domain[8]. This is a radically different approach compared to conventional systems that use sensors that first convert the physical stimuli to electrical signals and then perform the processing using analog or digital electronic computers.

Our work provides strong evidence that machine learning with in-sensor computing is a viable technological option for real-time, secure, and highly efficient wearable devices and edge computing devices. We present a demonstration of a complete sensor with built-in machine-learning capabilities via a wearable accelerometer that uses neuromorphic

[1]Institut Interdisciplinaire d'Innovation Technologique, Université de Sherbrooke, Sherbrooke, Canada. [2]Department of Kinesiology, Faculty of Medicine, Interdisciplinary Center for Research in Rehabilitation and Social Integration (Cirris), Université Laval, Québec, Canada. ✉e-mail: julien.sylvestre@usherbrooke.ca

computing in the mechanical domain to detect subtle gait patterns in human subjects. Our device implements its sensing and computing functions using the displacements of mechanical structures, and as a result, it can perform machine learning on sensor data in an especially efficient manner. The device is attached to the left foot, and is able to detect in real-time changes in the gait far better than a linear signal classifier. Unlike conventional benchmark tasks[9], this gait analysis task provides a difficult real-world testbed to analyze the benefits of in-sensor computing, as it requires a fully-integrated wearable device that must also be robust against variability in the gait accelerations, walking speed, morphology of the test subjects, and noise and non-ideal sensor characteristics.

Our fully integrated device allows a direct comparison of in-sensor computing with a conventional solution (i.e., machine learning software executing on a commercial microcontroller). Building on previous work, our device uses a custom micro-electromechanical system (MEMS) accelerometer that couples the inertial movement of a suspended proof mass (for acceleration sensing) to the nonlinear oscillations of a doubly clamped beam (for computation)[10,11]. Here, the MEMS device is additionally shielded against mechanical and electromagnetic perturbations, thermally stabilized, and, most importantly, tightly packaged with all its auxiliary electronics so that it can be tested as a miniaturized wearable device. As it is commonly done in the field of *physical reservoir computing*[12,13], the auxiliary electronic components are used to drive our device and to increase the dimensionality of its acceleration data representation using a feedback mechanism.

Our prototype device is shown to consume power at a level that is similar to a heavily optimized commercial micro-controller implementing the same gait classification task in software (see the section "In-sensor computing for wearable devices"). The total power consumed by the MEMS prototype device was measured and compared to a model summing the calculated power consumed by each subsystem ($970 \pm 10$ mW total measured power, 958 mW calculated, see the "Methods" subsection "Power consumption"). This analysis showed that a simple re-design of certain subsystems in our prototype device would further reduce its power consumption by an order of magnitude (down to an estimated 94 mW, see "Methods" subsection "Power consumption"), thus providing a significant advantage over conventional microcontroller-based solutions and paving the way for wearable devices that are much smaller or function much longer on a battery charge. Our results thus firmly establish in-sensor computing in MEMS accelerometers as a competitive alternative to conventional embedded software solutions in a device that is integrated and effective enough for a gait classification task that is difficult and relevant for real-world applications.

In addition, only the classification labels are transmitted (wirelessly) by our in-sensor computing device. The raw physiological data (i.e., the accelerations) are nonlinearly transformed through the dynamics of the MEMS before they are digitized. The digital data in the electronics feedback mechanism cannot be inverted to retrieve the raw accelerations because of the complexity of the dynamics of the MEMS, thus providing built-in data security. This feature of in-sensor computing could be especially relevant for wearable medical devices to protect the privacy of their users and to save energy by reducing the amount of transmitted data and the use of encryption algorithms.

The most fundamental challenge for in-sensor computing has arguably been the development of an appropriate coupling between the computing functions and the sensor functions of a device. Our work shows that this coupling is conceptually simpler to implement in a device when the computing functions are realized using degrees of freedom that can also be influenced by the physical stimuli that are being measured. We achieve this by coupling a nonlinear resonator to a suspended-proof mass in a microfabricated accelerometer. Data processing is realized in the acceleration sensor by leveraging the computing model known as *reservoir computing*[14], which has been tremendously influential in the development of computing systems that use the measurable degrees of freedom of physical dynamical systems (aka. physical reservoir computers[9]). Recent examples of physical

reservoir computers with in-sensor computing capabilities have included the coupling of radio-frequency electromagnetic waves to electron-spin reservoir computers[15] and a light-sensitive memristor reservoir computer[16]. Our work advances the nascent field of in-sensor computing by demonstrating the energy efficiency, compactness, and data processing capabilities of machine learning concepts implemented in the same physical domain as the sensory information in a highly functional wearable device.

## Results

### MEMS device with sensing and trainable computing capabilities

We address a human gait classification task (Fig. 1a–c) with a microelectromechanical (MEMS) accelerometer that performs both sensing and computing in the mechanical domain. The mechanical reservoir computer is implemented using the dynamical nonlinearity of a thin silicon beam clamped at both ends, which is similar to the device described previously[10]. When the beam is driven by an oscillating electrostatic force that is sufficiently large, the amplitude of its oscillations exhibits complex nonlinear dynamics (Fig. 2a, b), that are used for reservoir computing. The driving electrostatic force on the clamped beam is applied through a suspended proof mass that moves when accelerations are applied to the device (Figs. 1a and 2d), as described previously[11]. The amplitude of the driving force varies with the distance between the proof mass and the equilibrium position of the beam, and the parameters of the system are chosen so that the amplitude of the oscillations of the beam is a complex nonlinear function of the position of the proof mass (and therefore, of the accelerations, see Fig. 1b). In order to increase the complexity of the signals that can be generated from the amplitude response of the beam, a feedback technique[12] is employed to create multiple different virtual responses using a time-multiplexing technique (Fig. 1b), each being a different nonlinear function of the accelerations. The virtual responses are sampled at regular intervals with conventional electronics to generate a vector of 'activation values' at each time interval (a complete block diagram of the system is shown in Supplementary Fig. 2). The scalar product between this vector and a trained weights vector is finally computed in a conventional microcontroller at each time interval to produce the output classification for the type of gait (Fig. 1b).

The feedback approach to generate virtual activation values has been used successfully in a wide variety of physical reservoir computers[9], as it allows to adjust the memory of a physical reservoir (which is often fixed by the hardware) to the memory required to solve a task[10,17], and as it allows to leverage the fast or energy-efficient physical reservoirs to perform complex nonlinear computations, that are then efficiently processed by conventional electronics. As we demonstrate in this work, even sub-optimized physical reservoir computers that rely on a feedback technique can achieve performance levels that are competitive with conventional electronics[18–20].

A different weight vector is used for each of the two gait patterns (toe-out, TO, and trunk-lean, TL, Fig. 1b), so the device can discriminate between a normal gait (N) or an abnormal gait with either (TO or TL) or both (TOTL) of the TO and TL patterns. The MEMS device was packaged and integrated on a small printed circuit board (Fig. 2e) so that it could be attached to a shoe with minimal interference to the gait. Additional details about the MEMS device and the weights training process are provided in the "Methods" subsections "MEMS design and fabrication" and "MEMS packaging and board integration". Similar MEMS devices have also been described elsewhere[21–27].

For our MEMS device, the in-sensor computing that is done in the mechanical domain is the result of the modulation of the high-frequency driving voltage (around 250 kHz) that is the input to the beam reservoir computer by the low-frequency displacement (below 400 Hz) of the proof mass, which is sensitive to external accelerations. As the electrostatic driving force is proportional to the square of the drive voltage, the beam is driven into fast in-plane oscillations (around 500 kHz). As displayed in Fig. 1b and Supplementary Fig. 1, gait acceleration signals have a low-frequency content (0.5–15 Hz) and small amplitude (~1 g) as a result of the biomechanics of the human body[28,29]. The inertial sensing component of our MEMS system is designed to respond to these acceleration signals (see the sensitivity curve in

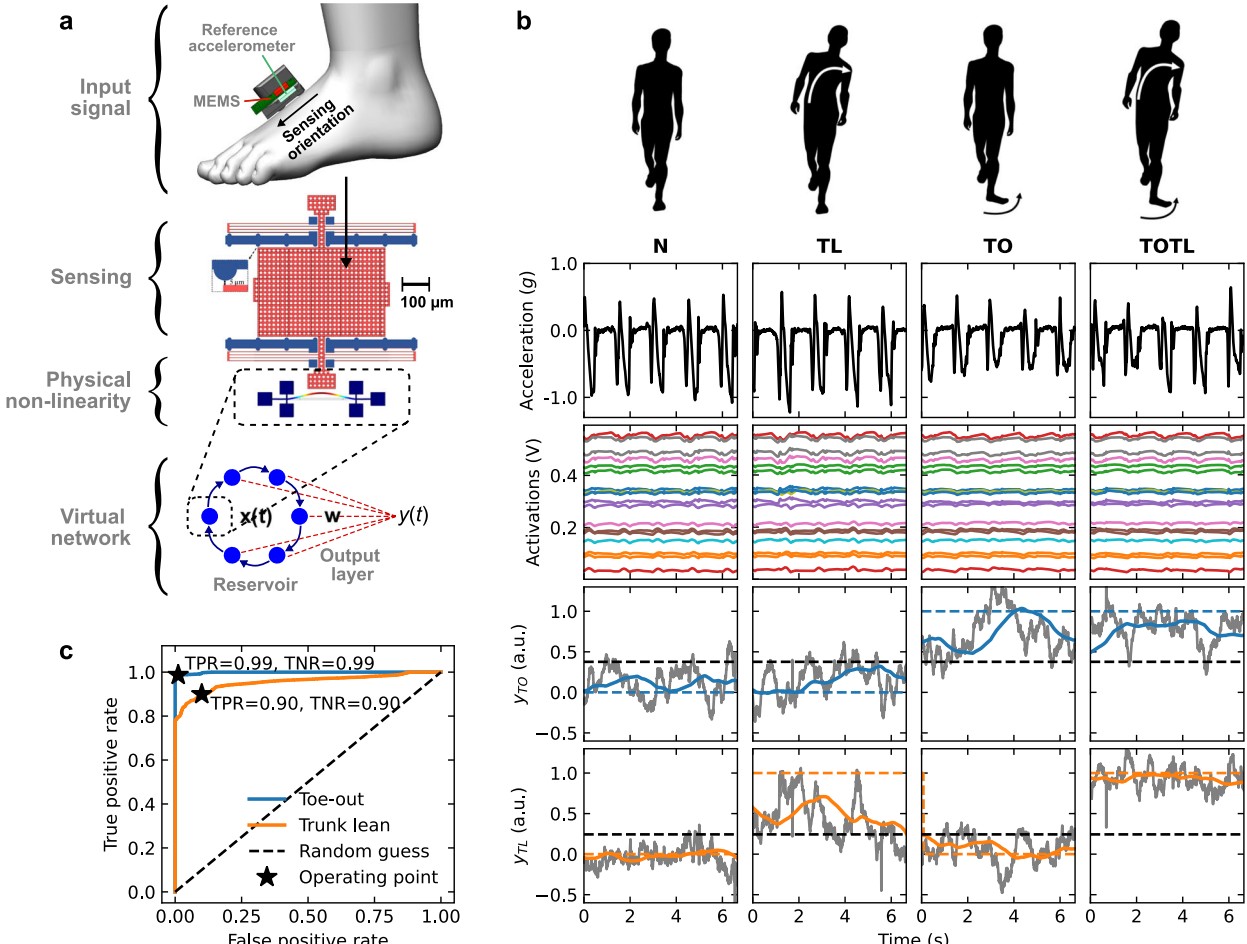

**Fig. 1 | General description of the MEMS gait analysis system. a** Schematic description of the MEMS gait analysis system, showing the position of the MEMS and reference accelerometers on the foot, the MEMS accelerometer sensing element coupled to the non-linear beam resonator, and the network of virtual activations used to generate the output signal $y(t)$. **b** Schematic representation of the gait patterns (top row), accelerations measured by the reference accelerometer (second row), virtual activations for 18 randomly selected nodes (third row), and gait discrimination output signals $y_{TO}$ for the toe-out gait and $y_{TL}$ for the trunk-lean gait

(fourth and fifth rows, respectively, gray for the raw signals, and color lines for the time-averaged signals), together with the threshold levels (fourth and fifth rows, black dashed lines) and detector binary outputs (fourth and fifth rows, dashed color lines). **c** Receiver operating characteristic curve for both walking patterns. MEMS micro-electromechanical system, N normal gait, TL trunk-lean gait, TO toe-out gait, TOTL toe-out and trunk-lean gait, a.u. arbitrary units, TPR true positive rate, TNR true negative rate.

Fig. 2d). By coupling the inertial proof mass to the driving force acting on the doubly clamped beam, efficient computing is achieved on a timescale set by the short ($\tau$ = 150 μs) decay time of the beam oscillator (Fig. 2a). The delayed feedback loop (details in section Supplementary Note 3) and leaky integrator (details in the section "Training the MEMS system") are used to increase this relatively short intrinsic memory, to match the frequency content of the gait acceleration signals. For the virtual network with $N = 100$ nodes that were used in this work, the envelope of the beam oscillation was sampled at a rate of $\theta^{-1} = 1/70$ MHz $> \tau^{-1}$ to produce the activation signals so that the raw classification signals were updated from the linear combination of the activation signals at a rate of 142.85 Hz. Since this update rate is too high for the frequency content of the biomechanical task, a moving average was performed over 300 time steps. This resulted in an effective computing timescale of ~2.1 s, which was well matched to the gait accelerations, and also filtered noise at frequencies higher than the bandwidth of the acceleration signals.

### In-sensor identification of human gait patterns
In-sensor computing with our MEMS device was applied to the detection of the four different gait patterns (N, TO, TL, TOTL) using a single device

attached to the left foot. The automated identification of these gait patterns is relevant clinically, as it may be used for the management of certain musculoskeletal diseases, such as knee osteoarthritis, for instance, via gait retraining using real-time biofeedback[30]. While gait pattern identification can be performed in laboratories equipped with 3D motion capture systems or using multiple inertial measurement units[31–33], the difficulty of the task is compounded by our use of a single accelerometer, which is motivated by our objective to develop the simplest, lowest cost, and most unintrusive wearable device.

In our study, ten healthy subjects were instructed to walk on a treadmill with the MEMS system attached to their left shoe while alternating between the four gait patterns (see the "Methods" subsection "Gait analysis protocol" and Supplementary Note 2 for details). Relatively slow walking speeds, between 0.36 and 0.72 m/s, were set on the treadmill to correspond to a range of speeds in a population with knee pain. This range (±33% around the median) is much larger than the expected range of natural variations for a given person (e.g., ±3%)[34]. The virtual activations from the MEMS reservoir computer and the accelerations measured by a conventional accelerometer co-located with the MEMS were sent to a computer in the laboratory. The training and the testing phases were performed

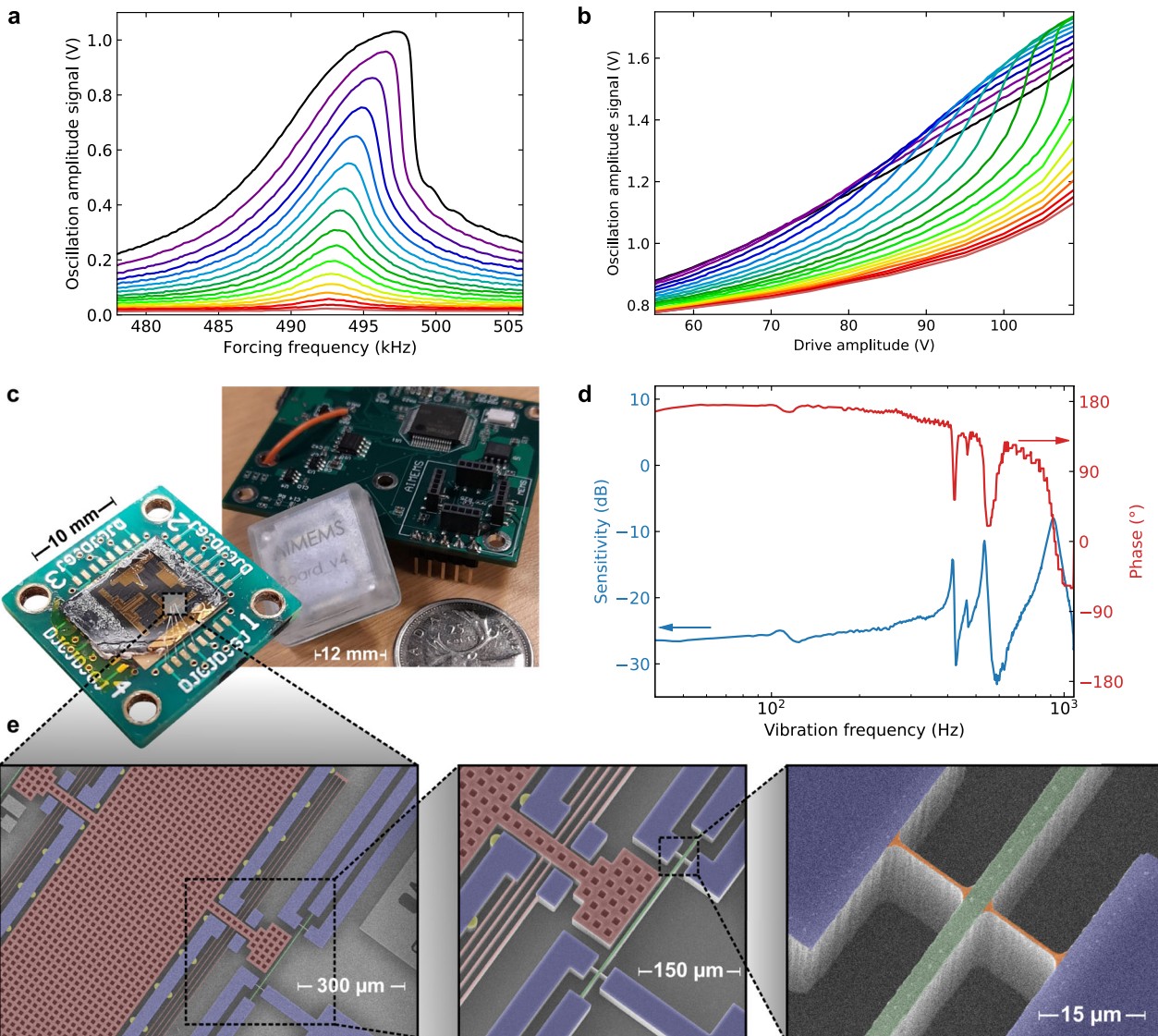

**Fig. 2 | Mechanical response of the gait analysis system. a** Frequency response curves for the silicon beam for a drive voltage amplitude that is increased from 30 V (red) to 109 V (black) in steps of 5 V (up to 105 V, black curve at 109 V). The Duffing stiffening nonlinearity of the silicon beam can be observed in the asymmetry of the frequency response curves. **b** Drive amplitude voltage sweeps for the silicon beam for a forcing frequency that is increased from 493.4 kHz (black curve) to 501.0 kHz (red curve) in steps of 400 Hz. These sweeps show that the deviation from the linear regime is increased as the drive voltage amplitude and forcing frequency are increased. **c** Photographs of the packaged MEMS and its main electronics board. **d** Inertial response of the MEMS, where the sensitivity is displayed as the amplitude of the oscillation signal (in V) per unit of acceleration applied (in standard gravity $g$), converted to decibels (dB) using a reference level of 1 V per $g$ (at 0 dB). **e** Manually colored scanning electron microscope images of a device (red: inertial mass and spring suspension, blue: anchors, green: clamped beam, yellow: inertial mass stoppers, orange: strain gages).

independently for each participant to obtain personalized classifiers with good performance. For each participant, we used a four-fold cross-validation (see the "Methods" subsection "Training the MEMS system" for details). For each of the four splits of the cross-validation, the training was performed on three folds using a ridge regression, and the testing was performed on the remaining fold. The classification performance was assessed by computing the average of the area under the curve (AUC) of the receiver operating characteristic (ROC) curve, computed over the four splits (an AUC of 0.5 corresponds to a random classifier while an AUC of 1.0 corresponds to a perfect classifier). See Fig. 1c for an example of a ROC curve and Fig. 3 for a characterization of the performance of the MEMS system. Figure 3 also presents for comparison the performance of two conventional software solutions (an echo-state network, ESN, and logistic regression, LR, details in the "Methods" subsection "ESN and logistic regression") that were

not implemented using in-sensor computing and that instead used the data acquired from the conventional accelerometer. The weight vectors obtained in training could finally be transferred back to the MEMS system, which was then used to classify gait patterns in real-time. The real-time performance of the system was then evaluated while a subject was walking on the treadmill with different gait patterns (see Supplementary Movie 1 for a demonstration).

Our results show that the TO and the TL gait detectors could both perform very well, with a typical probability for correctly identifying the TO pattern while the subject was walking with that pattern (TPR) above 99% and a typical probability for incorrectly identifying the TO pattern while the subject was not walking with that pattern (FPR) below 1% (respectively, 90% and 10% for the harder TL pattern, see Fig. 1c). This level of performance for training customized to a single subject could be useful in the context of

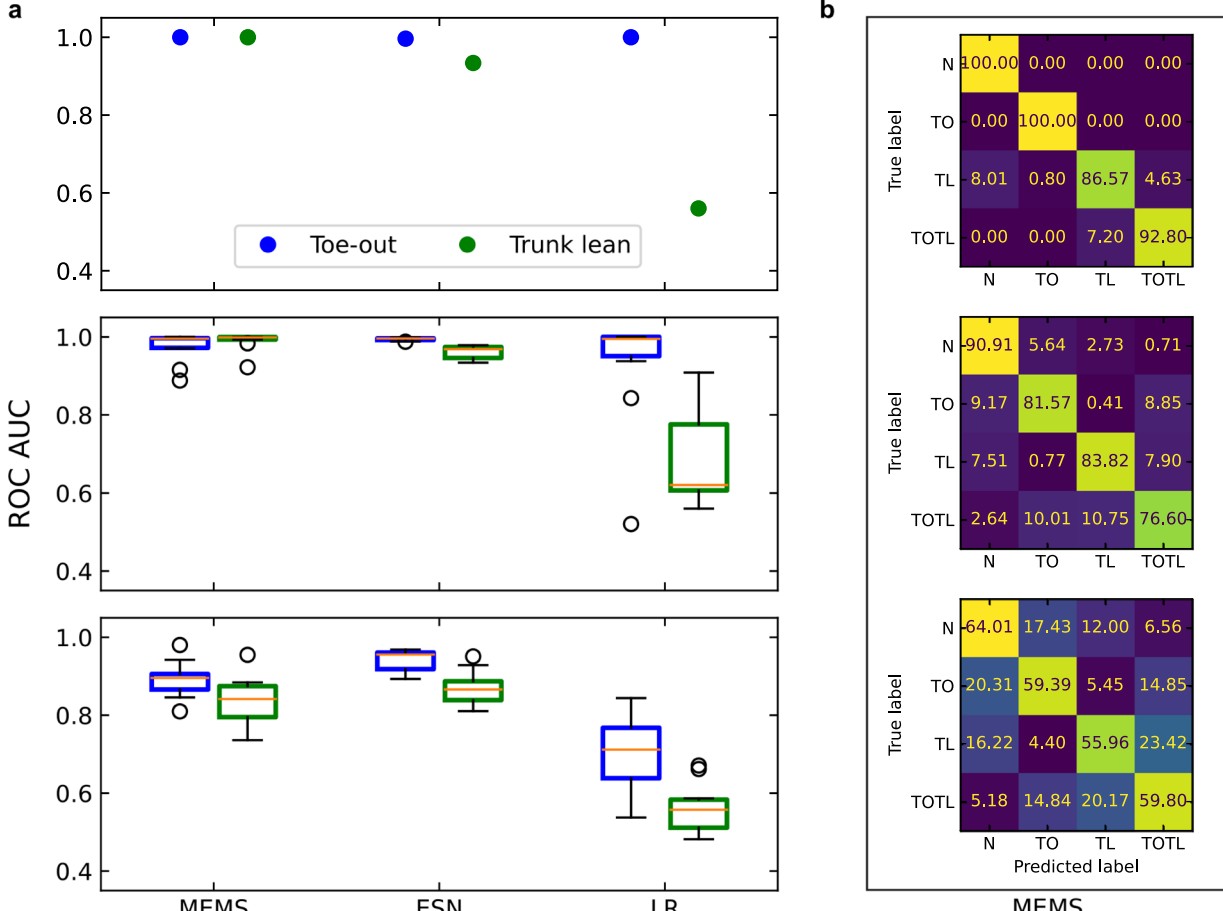

**Fig. 3 | Performance of in-sensor classification. a** Areas under the receiver characteristic curve for the toe-out (blue) and trunk-lean (green) gaits, for the MEMS, ESN, and LR classifiers, for a single participant walking at a steady pace (top panel), for all participants walking at a single speed of 0.63 m/s (middle panel), and for all participants, all walking at five different speeds (bottom panel). The box plots displayed in the bottom two panels show the first and third quartiles (boxes), the median (orange lines), 1.5 times the interquartile range (whiskers), and the data points outside the whiskers (circles). **b** Corresponding confusion matrices for the MEMS classifier showing the probability (in percentages) of the detector to predict a gait pattern (columns) for a prescribed gait pattern (rows). MEMS micro-electromechanical system, ESN echo-state network, LR logistic regression, ROC receiver operating characteristic, AUC area under the curve, N normal gait, TL trunk-lean gait, TO toe-out gait, TOTL toe-out and trunk-lean gait.

personalized medicine[35], where a device could be trained in a clinical environment to provide a high-accuracy gait classifier to achieve a specific therapeutic objective. Figure 3a further shows the AUC for subtasks with various levels of complexity. It can be observed that the MEMS classifications were robust against variations in the gait patterns across individual subjects when they all walked at the same speed, even if there were substantial differences within the group of subjects (10 participants, age 33 ± 15 years, height 173 ± 8 cm and weight 75 ± 13 kg, reported as mean ± standard deviation). When the difficulty of the gait identification task was further increased by using a single set of weight vectors for all walking speeds, the performance for the MEMS classifications was reduced (median AUC of 90% for TO and 84% for TL), principally due to an increase in the frequency of errors between the normal and TO patterns, and between the TL and TOTL patterns (Fig. 3b).

This level of classification performance with the MEMS device is very good compared to what can be achieved with linear classifiers, especially for the harder TL identification task. Figure 3a shows, for instance, that the median AUC for the TL pattern with all participants and all speeds is much better with the MEMS device (84%) than with a logistic regression classifier that did not perform significantly better than random guessing (median AUC of 56%). A similar logistic regression classifier could perform well for TO identification at a fixed speed, but its performance was much lower than

the performance of the MEMS device on all other tasks (TL and TO with combined speeds). This establishes that only nonlinear classifiers can perform well under the conditions of our gait identification task and further illustrates the usefulness of sophisticated in-sensor processing for this application.

**In-sensor computing for wearable devices**

In order to have a relevant technological impact, in-sensor computing devices need to generalize while performing complex computations, i.e., they must be robust against variations in the inputs that were not seen during training, as well as a certain level of noise. This robustness is a hallmark of neuromorphic computing techniques, including reservoir computing. In addition, these techniques can often model the nonlinearities in sensor systems[36–38], so they perform well even with non-ideal sensors. This could be leveraged to alleviate the design requirements of many sensors, thus potentially lowering their cost of fabrication or calibration.

The performance of our prototype device as a pure sensor (Fig. 2d) was far from the state-of-the-art for MEMS accelerometers, with a sensitivity between 0.05 V per *g* and 0.1 V per *g* in a bandwidth of 400 Hz, some in-band resonances and other problems (see Supplementary Note 1 for details). Nevertheless, the beam reservoir computer was able to learn these performance limitations, to classify gait patterns as well as a neural network

operating on data from a commercial accelerometer. Figure 3a shows a comparison of the classification performance of the MEMS device and of a conventional system built using a commercial accelerometer and software ESN[39] executed on a microprocessor. It can be observed that the MEMS device and the ESN both learned the variability between subjects and speeds to achieve similar classification performance and to perform better than the linear classifier. The MEMS device further had to learn the non-ideal behavior of its acceleration sensor so that it was arguably solving a more difficult task. It could, of course, be expected that more elaborate classifier software could achieve better performance with the commercial accelerometer data; the comparison was performed here with an ESN with optimized hyperparameters and with the same number of reservoir nodes as the number of virtual activations in the MEMS device (see the "Methods" subsection "ESN and logistic regression" for details).

This demonstration that the MEMS solves the gait classification task, as well as an ESN, further allows a direct comparison of the energy consumed by our system and by a conventional electronics system. The MEMS was packaged on a small electronic board that was used to implement some non-essential functions such as data logging, subsystem control, wireless communication, voltage level conversion, and parameter tuning for the signal processing chain. It was also used to implement the functions that are essential for the current MEMS, including the battery, a high voltage drive with an amplitude modulator, an analog-to-digital converter, a microcontroller implementing a buffer of activation values, and a scalar product with a weight vector, a digital-to-analog converter, and a signal processing chain to measure the position of the beam. With all functions active, the power consumed by our prototype system was measured to be $970 \pm 10$ mW. A system executing the ESN on a state-of-the-art microcontroller and using data from a commercial accelerometer was measured to similarly consume $280 \pm 40$ mW of power. By breaking down the power consumption of our prototype by subsystems, eliminating the non-essential functions, and improving some simple elements of the MEMS design, we can calculate that a better-optimized version of our system could consume only 94 mW of power, already much better than the heavily optimized ESN system built with commercial components. More complex design modifications could further reduce the power consumption below 12 mW, for instance, by co-integrating the control electronics with the MEMS[40] or increasing the quality factor of the MEMS resonator with vacuum packaging[41] (see the "Methods" subsection "Power consumption" for details). The lowest power consumption could arguably be achieved by eliminating the feedback circuitry to create activations entirely in the mechanical domain and by forming the linear combination of the activations in the analog domain, as discussed in the "Methods" subsection "Power consumption".

## Discussion

We have described a device that is both an acceleration sensor and a neuromorphic computer and that performs its sensing and nonlinear computing functions in the mechanical domain via the displacement of suspended microstructures. This wearable device was shown to successfully implement the concept of in-sensor computing for the difficult task of identifying the gait pattern used by a human subject, using only the acceleration measured on one foot. The most important characteristics of the device that enabled this successful demonstration were its computing capabilities (linear classifiers cannot solve the gait classification task), robustness against data variations and non-ideal sensor behavior, small size, and low power consumption. These characteristics are natural benefits of the co-integration of the sensing and neuromorphic computing functions in a single device. Another benefit of in-sensor computing is that the sensor data never leave the device or are never actually converted to the electrical or digital domains, thus offering a high level of privacy that could be especially relevant for medical devices.

These benefits that are important for medical devices are also highly desirable for edge computing devices in applications for the Internet of Things. In these applications, in-sensor computing could further allow the amount of data transmitted by the edge sensors to be drastically reduced, from the full (compressed) bandwidth at the Nyquist frequency of the signal down to the rate of identification of relevant features in the signal. In addition to alleviating the data congestion issues that are typical of Internet of Things applications[42], this could greatly increase the battery life of edge sensors, thus facilitating their field deployment at a large scale. Our demonstration of a single device that solves a complex real-world task by performing both sensing and neuromorphic computing functions directly in the mechanical domain can thus be considered an important milestone for the broad deployment of sensors and machine learning capabilities in emerging applications for wearable medical devices and for the Internet of Things.

## Methods
### MEMS design and fabrication
The suspended inertial mass and doubly clamped beam were defined by photolithographically patterning their shapes in the AZ MIR 701 photoresist, deep reactive-ion etching (Bosch process) of the 50 μm device layer of a P-type (boron dopant, 0.02 Ω cm) silicon on insulator (SOI) substrate, and then releasing by HF vapor etching of the 4 μm buried oxide (BOX) through perforations in the proof mass (10 μm-side squares on a pitch of 20 μm), while their anchors (minimum width of 35 μm) remained attached to the 350 μm-thick handle layer due to their larger surface area and absence of perforations. The device was finalized by a metallization step where a stainless steel laser-cut stencil attached over the die allowed the deposition of a Cr–Au film over the electrical traces using electron-beam evaporation. More details of the fabrication process are given elsewhere[11].

The 50 μm thickness of the device layer provided sufficient out-of-plane stiffness to prevent the pull-in of the inertial mass towards the handle across the 4 μm air gap left by the release step (etching of the BOX), but more importantly, provided substantial mass (49 μg) in a small footprint (1 mm$^2$), thus providing a sufficient sensitivity to uniaxial accelerations. Four pairs of 3 μm-wide folded flexure springs (436 and 488 μm long segments) were used to suspend the inertial mass. They were tuned to provide sufficient longitudinal displacement in reaction to accelerations (6.8 N/m spring constant) while restricting transverse and rotational displacements. An 8 μm-wide electrostatic transduction gap separated the inertial mass from the beam oscillator. The beam width (4 μm) and length (300 μm) were adjusted[10,11] for fast oscillations (much above the resonance frequency of the inertial mass, in a range where the resonator is not sensitive to inertial forces), as well as to easily achieve nonlinear dynamics through mechanical stiffening at large displacements. Thin 1.5 μm × 12 μm piezoresistive strain gauge pairs attached to the beam were included in the design to allow the differential piezoresistive measurement of its oscillations. Sets of stopper structures (18 μm diameter half-discs) surrounded the inertial mass in order to limit its range of motion to 5 μm, to protect the device from shocks and electrostatic pull-in, which could otherwise damage the device by short-circuiting the capacitive transduction gap.

### MEMS packaging and board integration
A schematic description of the system is shown in Fig. 4. The beam oscillation amplitude signals were preprocessed by an analog measurement circuit before being sampled by the analog-to-digital converter. The digitized samples were then fed to a leaky integrator which outputs the virtual node activations that were used during inference by the output layer, which was trained using a ridge regression. The activations were also delayed and fed back to the MEMS through the high-voltage drive circuit driven by a digital-to-analog converter. The detected gait pattern was obtained by thresholding the temporally averaged linear combinations of the activations.

The MEMS die was wire bonded to a 20 mm × 20 mm printed circuit board (PCB) chip carrier, and a plastic cover was attached to the PCB in order to protect the device from its environment (see Fig. 2c for a photograph of the packaged MEMS). Fine-pitch male headers soldered on the flip

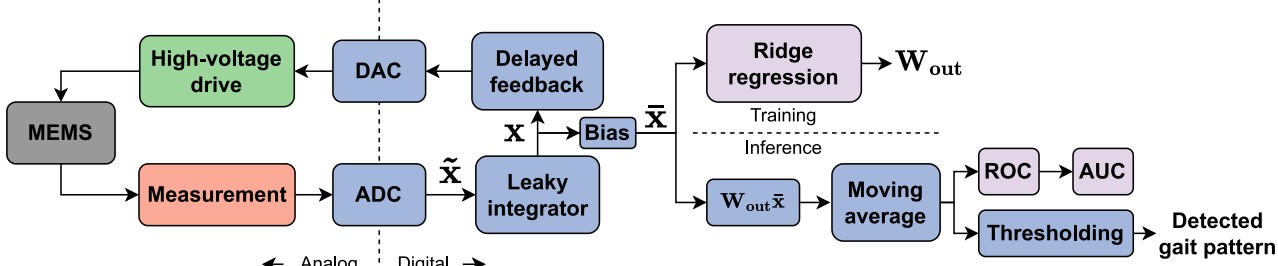

**Fig. 4 | Detecting gait patterns from MEMS virtual node activations.** Schematic description of the gait classification system. ADC analog-to-digital converter, AUC area under the curve, DAC digital-to-analog converter, MEMS micro-electromechanical system, ROC receiver operating characteristic, $\tilde{x}$ digitized samples; $x$ virtual node activations, $\bar{x}$ activation vector augmented with a constant bias term, $W_{out}$ output weight matrix.

side of the chip carrier PCB allowed for its mechanical and electrical interfacing to the control electronics board (see Supplementary Note 3 for a detailed description of the electronic modules) while matching mounting holes allowed the stacking of a reference accelerometer for characterization purposes. After mounting the packaged MEMS device, control electronics, and batteries in a 3D-printed wearable plastic enclosure, the enclosure was shielded with copper tape over the areas containing sensitive or high voltage signals for safety and to improve the integrity of electrical signals (see Supplementary Note 4 for details).

### Gait analysis protocol
Ten healthy adults participated in the study. All participants gave written informed consent to participate in the study, according to experimental procedures approved by the IRDPQ-CIUSSS-CP research ethical committee (2021-2269).

The participants were instructed to walk on a treadmill (SuperFit 2.25HP). The participants were equipped with a 3D-printed enclosure attached to the top of the left shoe. The enclosure included the MEMS device as well as a commercial reference accelerometer (Analog Devices ADXL326 accelerometer, ± 16×$g$, bandwidth of 0.5–1600 Hz, sensitivity of 51 mV per $g$). The signals produced by the MEMS and the reference accelerometer were both sampled at 14285 Hz by a National Instruments PCIe-6374 16-bit data acquisition card using shielded coaxial cables for electrical connections. The reference acceleration signal was then downsampled by a factor of 100 after applying a linear-phase, finite impulse response low-pass filter at the Nyquist frequency. The cables were attached to the lower leg of the participant up to the knee and were fixed on the treadmill with enough slack to let the participants walk comfortably.

After explaining the gait patterns to the participants, participants walked with each pattern for 15 s, for every treadmill speed considered in the study, from slow to fast, while feedback was given by a supervisor. For the trunk lean pattern, participants were instructed to stand on the left foot only to balance their trunk to keep this posture and then to apply this posture for every step taken by the left leg. For the toe-out pattern, they were required to angle their left foot outwards at an angle that was comfortable for them. Once their gait patterns were deemed stable by the supervisor, the participants each walked 20 sequences of 90 s duration, with different gait patterns and speeds. The first, second, third, and fourth sequences were in the normal (N), toe-out (TO), trunk-lean (TL), and toe-out+trunk-lean (TOTL) gait patterns, respectively, at a fixed speed of 0.36 m/s. Four additional sequences in the N, TO, TL, and TOTL gaits were then sequentially obtained at speeds of 0.45, 0.54, 0.63, and 0.72 m/s. For each recording, the first 10 seconds of data were discarded (this 10 s segment served to give enough time to the participants to adopt each gait pattern). If the supervisor judged that the gait pattern was not respected for the full duration of a given sequence, that sequence was discarded and reacquired.

### Training the MEMS system
As shown in Fig. 4, the virtual node activations $x$ used by the output layer to classify gait patterns in real-time were obtained through leaky integration of

the beam oscillation envelope samples $\tilde{x}$ :

$$x(n) = (1 - \alpha)x(n - 1) + \alpha\tilde{x}(n), \tag{1}$$

where $\alpha$ is the leaking rate and $\tilde{x}(n)$ is a vector containing $N = 100$ successively digitized samples at timestep $n$ of the detector output time series, with each sample corresponding to a different virtual node. In a training phase, these activations, which are non-linear transformations of the physical acceleration signals, were input to a ridge regression in order to obtain the output layer weights $W_{out}$. These weights were then used, in inference mode, as coefficients of linear combinations of the virtual node activations. A moving average of over 300 timesteps was applied to the result of the linear combinations to produce the detection signal. The moving average and leaky integration increase the short-term memory of the system, which has been shown to be useful for motion classification in other in-sensor systems[43]. This detection signal was thresholded at different levels to characterize the detector through its ROC, which in turn allowed the selection of a suitable threshold, chosen as the value that minimized the difference |TPR−TNR|. This threshold was finally applied during the test phase to output one of four possible classes (N, TO, TL, TOTL) for each timestep and subsequently construct the confusion matrices of Fig. 3.

During the training phase, the activations of the virtual nodes of the MEMS reservoir computer were stored after an initial 1000 timesteps transient was discarded, in a matrix $X \in \mathbb{R}^{(1+N) \times M}$, with $M$ the number of timesteps (an extra row of ones was added to have a constant bias in addition to the activation values). Based on the gait pattern prescribed to the participant, a target output matrix $Y \in \mathbb{R}^{2 \times M}$ was constructed. The first and second rows corresponded to a one-hot encoding for TO and TL, respectively. At timestep $n$, the $n$th column of $Y$ thus corresponded to the N, TO, TL or TOTL gait pattern for the patterns [0, 0], [1, 0], [0, 1] and [1, 1], respectively. An output weight matrix $W_{out} \in \mathbb{R}^{2 \times (1+N)}$ was obtained by ridge regression,

$$W_{out} = YX^{T}(XX^{T} + \beta I)^{-1}, \tag{2}$$

where $\beta$ is a regularization parameter used to control overfitting, and $I$ is the identity matrix of size $N + 1$. To use the output weight matrix $W_{out}$ for the classification of the walking pattern at the timestep $n$, the matrix–vector product $W_{out}\bar{x}(n)$ was computed with the activation values $x(n)$ at time step $n$ and a constant term for the bias, arranged in a single vector $\bar{x}(n)$. Finally, some of the parameters of the MEMS system had to be tuned to obtain good performance. The tuning procedure for these MEMS hyperparameters is described in Supplementary Note 5.

A $k$-fold cross-validation procedure was used for training and testing. Each recording was split into four folds. Three folds were used for training, and the remaining fold was used for testing. The procedure was repeated four times (resulting in four splits), each time using a different fold for the testing so that each fold was used exactly once for the testing. In all splits, the testing fold was never used during the training phase. All of the reported classification performance metrics were averaged over the four splits.

### ESN and logistic regression

The ESN was implemented in the software. It used the single-channel acceleration signal $u(n)$ from the reference accelerometer as input and produced a state $\boldsymbol{x}(n)$ according to

$$\tilde{\boldsymbol{x}} = \tanh\left(\boldsymbol{W}_{\text{in}}\begin{bmatrix}1\\u(n)\end{bmatrix} + \boldsymbol{W}_{\text{r}}\,\boldsymbol{x}(n-1)\right) \tag{3}$$

$$\boldsymbol{x}(n) = (1-\alpha)\boldsymbol{x}(n-1) + \alpha\tilde{\boldsymbol{x}}(n), \tag{4}$$

where the hyperparameter $\alpha$ is the leaking rate, $\boldsymbol{W}_{\text{in}} \in \mathbb{R}^{N \times 2}$ is the input weight matrix, and $\boldsymbol{W}_{\text{r}} \in \mathbb{R}^{N \times N}$ is the reservoir weight matrix. Both $\boldsymbol{W}_{\text{in}}$ and $\boldsymbol{W}_{\text{r}}$ were initialized randomly from uniform distributions (between $-1$ and $+1$), then sparsified by randomly replacing entries by 0 (with a probability given by the hyperparameter called the *sparsity probability*) and finally scaled by multiplying by a scaling hyperparameter (for $\boldsymbol{W}_{\text{in}}$, one scaling hyperparameter was used for the input and another was used for the bias). The ESN hyperparameters were taken from an earlier study on a gait event detection task[44] and are given in Supplementary Note 6. In that study, the hyperparameters with the largest influence on the performance (the input scaling, the bias scaling, the spectral radius, and the leaking rate) were optimized by the CHARC method[45], and the other hyperparameters were taken from previous studies and not optimized further since they did not have much influence on the performance. In the CHARC method, the hyperparameters were selected to maximize the ratio between the kernel rank (which measures the ability of the ESN to distinguish distinct inputs) and the generalization rank (which must be decreased to improve the ability of the ESN to recognize similar inputs). After discarding the initial 1000 timesteps to account for the initial ESN transient, the states from a recording with $M$ timesteps were assembled in a matrix $\boldsymbol{X} \in \mathbb{R}^{(2+N) \times M}$, together with a row of ones to add a bias term and a row corresponding to the acceleration signal. The rest of the procedure was then the same as for the MEMS system: for each split of the cross-validation, a matrix $\boldsymbol{W}_{\text{out}}$ was obtained by ridge regression, predictions were generated using the matrix-vector product of the state-bias-acceleration vector with $\boldsymbol{W}_{\text{out}}$ and a moving average of 300 timesteps was applied.

For the logistic regression, a digital low-pass filter (FIR filter with a Hamming window, cutoff frequency of 71.5 Hz, order 40) was applied to the acceleration signal from the reference accelerometer. The first 1000 acceleration values were removed in order to keep the same training data length as for the MEMS and ESN methods. Then, at each timestep $n$, the last 715 acceleration values (a window corresponding to the last 5 s) were down-sampled by a factor of 2, and the resulting 358 acceleration values, as well as the mean of the last 715 points, were used as features for the logistic regression. These features were standardized by subtracting their means and by dividing by their standard deviations, with the means and standard deviations obtained from the training dataset. A two-row matrix $\boldsymbol{Y}_{\text{p}}$ containing the probabilities for the TO and TL classes was calculated by applying the logistic function to a linear combination of the features plus a bias term. The parameters of the linear combination and the bias were obtained during training by a coordinate descent algorithm implemented in the liblinear solver in the scikit-learn package[46]. In this algorithm, the regularization parameter $C$ was set to 1.0, and the maximum number of iterations was 100. The same procedure as for the MEMS and the ESN was then followed (for each split of the cross-validation, $\boldsymbol{Y}_{\text{p}}$ was computed, a moving average over 300 timesteps was applied, and performance was evaluated).

### Power consumption

In order to compare the power consumption of the MEMS system to a conventional implementation that uses separate sensor and processing components, the ESN described in section M4 was also implemented on an Adafruit Industries HUZZAH32 Feather board, that is built around an Espressif Systems ESP32 microcontroller. The board was connected to an STMicroelectronics ISM330DHCX inertial measurement unit. The WiFi,

radio, and Bluetooth functions were deactivated to save power. Power was supplied through the board USB connector, and everything was packaged in a 3D-printed plastic enclosure similar to the one of the MEMS system.

Power measurements were performed on our MEMS prototype as well as on the ESN running on the Feather system by connecting an ammeter in series with a 5 V benchtop power supply replacing the batteries or USB power of either system. Power dissipation was measured in inference mode, without data transmission. For the MEMS system, a simple LED board addressed by the microcontroller was used to provide real-time feedback about the detected gait pattern.

Current and projected energy requirements for the MEMS system were calculated by summing the power dissipation contribution of every electronic and electromechanical subsystem. These individual contributions, shown in Supplementary Table 3 and Supplementary Table 4, were obtained by first measuring the signal characteristics (root mean square (RMS) voltage ($V_{\text{out}}$) and dominant frequency) at the output of each active device that powers these subsystems. For operational amplifiers, the dominant frequency of signals was then used to compute equivalent load impedances ($Z_{\text{load}}$) from nominal values of passive load components. This in turn, allowed the conversion of the measured voltages into currents being sourced and sinked by the amplifiers ($I_{\text{load}}$). Finally, an estimate of power dissipation of all subcircuits ($P_{\text{total}}$) was obtained by adding the integrated circuit quiescent power dissipation ($P_{\text{quiescent}} = V_{\text{supply}}I_{\text{quiescent}}$, where $V_{\text{supply}}$ is the supply voltage difference, and $I_{\text{quiescent}}$ is the quiescent current, taken from the datasheet) to the power dissipated due to the application of a signal ($P_{\text{load}} = V_{\text{supply}}I_{\text{load}}/2$, since analog signals are bipolar and biased at 0 V). Power dissipation values for the microcontroller, the voltage-controlled oscillator (VCO) and the three voltage regulators were obtained from their respective datasheets. The power outputs of the voltage regulators, calculated using values from Supplementary Tables 3 and 4, were used to deduce their power dissipation through the efficiency curves obtained from their respective datasheets. The high-voltage amplifier power dissipation was deduced by disconnecting it from the system and by measuring the subsequent drop in the power supplied to the system.

A total power requirement of 958 mW was calculated with this method for the current version of the MEMS system (see Supplementary Table 3), consistently with the measured value of 970 ± 10 mW. This level of agreement validates our power analysis methodology, which thus provides a way to identify subsystems that can be optimized and to calculate expected power requirements for future designs. As detailed in Supplementary Table 4, a large reduction in power consumption could be achieved by simply changing the currently supplied voltages (+3.3, +5, and −5 V) to a single 2.5 V power rail, selecting lower power active chips and scaling component values accordingly, and eliminating unnecessary or data logging subcircuits such as the EEPROM chip, some voltage offset and pre-amplification stages. In addition, the multiplier integrated circuit, which dissipates an order of magnitude more power than other active components in our current implementation, could be replaced by a digital potentiometer controlling the gain of a voltage amplifier in order to implement amplitude modulation more efficiently. With these straightforward modifications, the power consumption would be reduced from its current value of 970 ± 10 mW (measured) down to 94 mW (calculated).

Lower power consumption levels could arguably be achieved by eliminating the feedback circuitry to create activations entirely in the mechanical domain. This could be achieved using multiple resonators[47] or multiple proof masses[48]. A completely mechanical implementation with the number of activations (approximately 100) and memory (on timescales on the order of seconds) required for complex time-series processing (such as gait analysis) has yet to be experimentally demonstrated, but hybrid systems with a few resonators or proof masses[25,49] could be a stepping stone toward this goal, that proportionally reduces the power consumption of the feedback electronics. Any fully integrated system should also have efficient drive and read-out mechanisms with efficient electrical implementations[50]. Finally, a complete MEMS system should implement a mechanism to adjust the weights in the linear combination of all its activation functions, to

support learning. It is unlikely that this could be achieved by tuning the structure of the MEMS, due to the economical costs of creating lithographic masks and, more fundamentally, to the inherent variability of the fabricated devices. An intriguing possibility would be to perform the linear combination on analog signals instead of in the digital domain, using an array of adaptable elements to adjust the weights, such as memristors[51]. The nonlinear processing of the physical signals could be performed efficiently by MEMS resonators with a high-quality factor, while low currents would flow from the memristor array into a summing amplifier with a high input impedance.

## Reporting summary

Further information on research design is available in the Nature Portfolio Reporting Summary linked to this article.

## Data availability

The recorded reference acceleration signals and the virtual activation signals for all participants, gait patterns, and walking speeds are available online[52].

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

## Acknowledgements

The authors thank Mathieu Bergeron, Cyril Bounakoff, and Adrien Merel for their help. J.S. and K.T. disclose support for the research of this work from the New Frontiers in Research Fund [grant number NFRFE-2018-00598].

## Author contributions

Guillaume Dion: Formal analysis, investigation, resources, software, writing—original draft, Albert Tessier-Poirier: Formal analysis, software, writing—original draft, Laurent Chiasson-Poirier: Formal analysis, investigation, software, Jean-François Morissette: Writing—review & editing, project administration, Guillaume Brassard: Software, Anthony Haman: Software, Katia Turcot: Funding acquisition, Julien Sylvestre: Conceptualization, writing—original draft, supervision, funding acquisition.

## Competing interests

The authors declare the following competing interests: a patent application was filed by Guillaume Dion and Julien Sylvestre for the MEMS technology used in this paper (WO/2020/124219). The other authors declare no competing interests.
