## [Peer Review File · Communications Engineering]

Reviewers' comments:

Reviewer #1 (Remarks to the Author):

In “In-sensor human gait analysis with machine learning in a wearable microfabricated accelerometer”, Guillaume Dion and co-authors present a study on a MEMS device with a suspended proof mass, couple the acceleration to the dynamics of oscillating beam, and create hardware reservoir computing (RC) implementations through the delay-based feedback approach. The authors demonstrated the MEMS device in a complete wearable system to identify the gait patterns of human subjects in real-time. In general, this paper is innovative and is of interest to potential readers working in the field of In-sensor computing. There are some comments on this work listed as follows:

Q1: In the manuscript, as described in Figure 4 and Figure S12, the delayed feedback signal is applied to the proof mass of the MEMS directly. But in the previous article which is mentioned in the manuscript as reference 13, the feedback signal is added to the preprocessed input to realize RC. In page 7 line 8, it said the “The delayed feedback loop and leaky integrator described in section M4 are used to increase this relatively short intrinsic memory, to match the frequency content of the gait acceleration signals.” It would be better if the author can described the mechanism and the advantages of this feedback approach in more detail.

Q2: As the author mentioned in fig. 2b, “the deviation from the linear regime is increased as the drive voltage amplitude and frequency are increased.” Does this phenomenon apply to drive frequency larger than 248.9 khz? And in S15 MEMS system hyperparameters part, the author chose fixed 249.4 kHz beam driving signal frequency as the system hyperparameters, whether the higher drive frequency the better, and how to determine an optimal choice.

Q3: In the manuscript, the page 8 line 28, the performance for the MEMS classifications will reduce if the walking speed is not fixed. In practice, the pace varies greatly. It would be better if the author can provide some suggestions to improve the robustness to wide walking speed range.

Q4: Can the devices be switched between nonvolatile and volatile types? If so, the authors are suggested to include these detailed information.

Q5: The short-term memory characteristics have been also used for in-sensor processing other physical stimulation, e.g., Nature Nanotechnology, 2023, 18, 882-888. The authors are suggested to refer to these prior arts.

Reviewer #2 (Remarks to the Author):

It is a detailed and complete work on the wearable microfabricated accelerometer with the capability of analyzing the human gait. But as a so-called in-sensor device, some problems are supposed to be elaborated in more detail. Although it is not a disruptive innovation considering your previous works, it is

an attractive and valuable work if the authors can consider the following comments.

1. In Figure 2b, I don't find the bifurcation points that often exist in nonlinear MEMS resonator. To the best of our knowledge, the bifurcation points and the nonlinear region are vital for MEMS-based reservoir computing.

2. As a so-called in-sensor device, I wonder that how you implement the experiment "in the mechanical domain" with DAC in real-time then? Did you provide a real-time processing in your system? According to your manuscript, I don't know the detail training data types, I guess it might be mechanical-to-digital that needs a DAC or ADC, so I think it is incapable of accounting for the real-time processing.

3. How many virtual nodes (activations in your manuscript) and how long delay length do you need for different tasks? Or for your providing task, what will they affect your results?

4. I notice the resonance is around 247kHz in Figure 2, but why did you use "a high-frequency driving force (around 500kHz)"? Please elaborate your tuning process of operation point and the influence of frequency on result in more details.

Reviewer #3 (Remarks to the Author):

This paper presents an incremental advancement of the author's previous work for using MEMS RC computing. Specifically, besides using this RC for gait classification, most of the technical work and demonstration and hardware were already demonstrated in the author's previous paper in the 2020 JMEMS journal. Below are the concerns the authors need to address:

1-The sentence "using displacement degrees of freedom that are in the same physical domain" in page 2 is not clear.

2-All raw physical data are processed in the mechanical domain, is not completely true. My understanding of the virtual neuron approach is that those samples need to be stored and processed with a microcontroller to provide the classification. Along this line how do you justify still using microcontrollers with this MEMS computing approach?

3-I am not sure the energy efficiency is a valid claim given as demonstrated in the paper a microcontroller consumes less power to perform similar classification without the RC approach.

4-In page 2, line 6 it refers to the sensitivity curve in Fig.2c, and nothing there is about that.

5-Not sure what the point of claiming 99% accuracy on page 9 is, if the individual classifier was trained for each user. To me, this sounds like a standard overfitting problem. Typically, you need to build a general model for all the users and then test it over them.

6-How do we justify the high required voltage for operation?

7-Authors need to acknowledge a large body of research on using MEMS for computing, one recent of which was published recently in the COMMSSENG journal where a MEMS computing approach was presented without the need for a microcontroller and thus achieving near zero power (Energy efficient integrated MEMS neural network for simultaneous sensing and computing).

We thank all reviewers for taking the time to read and formulate suggestions to improve our manuscript. Their expertise is greatly appreciated.

The reviewers' comments are copied verbatim below (italicized). Every specific comment or question was addressed with the modifications that are described below, and that are identified in the revised manuscript in green. Our references to page and line numbers are for the revised manuscript. Bibliographic reference numbers might be different in the revised manuscript, but are not necessarily highlighted in green.

In addition to addressing the reviewers' specific comments, we have tried to reformulate the third paragraph of the Introduction (in the original manuscript), to address some of the general comments of the reviewers, and to better articulate the elements of novelty in our manuscript. That portion of the Introduction now reads (page 2/line 23 to page 4/line 4):

“Our work provides strong evidence that machine learning with in-sensor computing is a viable technological option for real-time, secure and highly-efficient wearable devices and edge computing devices. We present the first demonstration of a complete sensor with built-in machine learning capabilities, via a wearable accelerometer that uses neuromorphic computing in the mechanical domain to detect subtle gait patterns in human subjects. Our device implements its sensing and computing functions using ~~displacement degrees of freedom that are in the same physical domain the displacements of mechanical structures,~~ and as a result ~~can~~ perform machine learning on sensor data in an especially efficient manner. The device is attached to the left foot, and is able to detect in real-time changes in the gait far better than a linear signal classifier, ~~especially when there is variability in the gait pattern and when the acceleration measurement is subjected to noise or non ideal sensor characteristics.~~ Unlike ~~conventional benchmark tasks (e.g., table 2 from reference 10), this gait analysis task provides a difficult real-world testbed to analyze the benefits of in-sensor computing, at it requires a fully-integrated wearable device, that must also be robust against significant variability in the gait accelerations, walking speed, morphology of the test subjects, and noise and non-ideal sensor characteristics.~~

~~Our fully integrated device allows a direct comparison of in-sensor computing with a conventional solution (i.e., machine learning software executing on a commercial microcontroller). Building on previous work, our device uses a custom MEMS accelerometer that couples the inertial movement of a suspended proof mass (for acceleration sensing) to the nonlinear oscillations of a doubly-clamped beam (for computation)^{13, 14}. Here, the MEMS device is additionally shielded against mechanical and electromagnetic perturbations, thermally stabilized and, most importantly, tightly packaged with all its auxiliary electronics, so that it can be tested as a miniaturized wearable device. As it is commonly done in the field of *physical reservoir computing* ^{12,13}, the auxiliary electronic components are used to drive our device and to increase the dimensionality of its acceleration data representation using a feedback mechanism.~~

Our prototype device is shown to consume ~~the same level of power~~ at a level that is similar to a heavily-optimized micro-controller chip implementing the same gait classification task in software. As shown below, a simple re-design of certain sub-systems in our prototype device would further reduce its power consumption by an order of magnitude, thus paving the way for wearable devices that are much smaller or function much longer on a battery charge (see section "In-sensor computing for wearable devices"). Our results thus firmly establish in-sensor computing in MEMS accelerometers as a competitive alternative to conventional embedded software solutions, in a device that is integrated and effective enough for a gait classification task that is difficult and relevant for real-world applications.

~~Importantly~~ In addition, only the classification labels are transmitted (wirelessly) by our in-sensor computing device. All the raw physiological data are processed in the mechanical domain within the device, where they are nonlinearly transformed through the dynamics of the MEMS, before they are digitized. The digital data in the electronics feedback mechanism cannot be inverted to retrieve the raw accelerations because of the complexity of the dynamics of the MEMS, thus providing built-in data security. This feature of in-sensor computing could be especially relevant for wearable medical devices, to protect the privacy of their users and to save energy by reducing the amount of transmitted data and the use of encryption algorithms."

Reviewer #1 (Remarks to the Author):

In “In-sensor human gait analysis with machine learning in a wearable microfabricated accelerometer”, Guillaume Dion and co-authors present a study on a MEMS device with a suspended proof mass, couple the acceleration to the dynamics of oscillating beam, and create hardware reservoir computing (RC) implementations through the delay-based feedback approach. The authors demonstrated the MEMS device in a complete wearable system to identify the gait patterns of human subjects in real-time. In general, this paper is innovative and is of interest to potential readers working in the field of In-sensor computing. There are some comments on this work listed as follows:

Q1: In the manuscript, as described in Figure 4 and Figure SI2, the delayed feedback signal is applied to the proof mass of the MEMS directly. But in the previous article which is mentioned in the manuscript as reference 13, the feedback signal is added to the preprocessed input to realize RC. In page 7 line 8, it said the “The delayed feedback loop and leaky integrator described in section M4 are used to increase this relatively short intrinsic memory, to match the frequency content of the gait acceleration signals.” It would be better if the author can described the mechanism and the advantages of this feedback approach in more detail.

In the reported MEMS device, the motion of the proof mass due to accelerations directly (physically) provides the input to the RC. This was clarified by moving the mention of reference 11 from the end of the paragraph to 4th sentence (page 5, line 10).

The reader’s attention is also better drawn to the explanations provided in section SI3 with the following modifications (page 6, lines 17-18):

“ The delayed feedback loop (details in section SI3) and leaky integrator described in section M4(details in section M4) are used to increase this relatively short intrinsic memory, to match the frequency content of the gait acceleration signals.”

The explanations provided in section SI3 were also improved (SI page 4, line 15):

“The delayed feedback signal was obtained by delaying the output of the leaky integrator (described in section M4) by 100 samples and scaling it by the feedback gain κ (see table SI1). This signal was superposed to the random binary mask signal and the result was output by the microcontroller onboard digital-to-analog converter (DAC) at 14285 Hz.”

We realize that the advantages of the feedback approach were not explained clearly enough, as the other reviewers have also commented on this. Please see the modifications to the manuscript that are described for Reviewer 2, Comment 2, and Reviewer 3, Comment 2.

Q2: As the author mentioned in fig. 2b, “the deviation from the linear regime is increased as the drive voltage amplitude and frequency are increased.” Does this phenomenon apply to drive frequency larger than 248.9 kHz? And in SI5 MEMS system hyperparameters part, the author chose fixed 249.4 kHz beam driving signal frequency as the system hyperparameters, whether the higher drive frequency the better, and how to determine an optimal choice.

The reviewer is right that in the original manuscript, figure 2b did not show the amplitude response around the forcing frequency of 498.8 kHz (driving voltage frequency of 249.4 kHz) given in table SI1. The amplitude response curves at driving voltage frequencies higher than 248.9 kHz were missing from the characterization data since the drive amplitude sweeps were not done when pull-in (for drive amplitudes higher than 115 V) could occur before the onset of bistability. However, the missing curves can be obtained by taking vertical slices (at fixed forcing frequencies) of the frequency response measurements (such as shown in figure 2a, but for decreasing frequency sweeps in order to show the right branch of the hysteresis loop, with the beam “jumping” from low to high oscillation amplitudes). A hybrid figure 2b was thus constructed to show a fuller range of forcing frequencies around the chosen 249.4 kHz driving voltage frequency. The updated figures 2a and 2b are now:

The caption of figure 2 was also updated:

“The Duffing stiffening nonlinearity of the silicon beam can be observed in the asymmetry of the frequency response curves (a), which increases as the drive voltage amplitude is increased from 30 V (red) to 109 V (black) in steps of 5 V (up to 105 V, black curve at 109V). Drive amplitude voltage sweeps (b) show that the deviation from the linear regime is increased as the drive voltage amplitude and forcing frequency (from 493.4 kHz for the black curve to 501.0 kHz for the red curve, in steps of 400 Hz) are increased.”

The discussion in section SI1 was updated to reflect this (SI page 1, lines 18-24):

“Using the same experimental setup as for the frequency sweeps, sweeping the driving signal amplitude (from low to high values) at fixed forcing frequencies from 493.4 kHz to 497.8 kHz in steps of 0.4 kHz produced the twelve topmost curves shown in fig. 2b. In order to show the response for a wider range of frequencies, the eight bottommost curves of fig. 2b (from 498.2 kHz to 501.0 kHz) were extracted from the frequency response data (for decreasing frequency

sweeps), since the response to drive amplitude sweeps was not measured for forcing frequencies over 497.8 kHz, for which pull-in of the inertial mass happened before the “jump” in oscillation amplitude could be observed.”

The reasons behind the choice of a driving frequency of 249.4 kHz were clarified in section SI5 (SI page 8/lines 18 to SI page 9/line 5):

“The beam driving signal frequency was fixed at 249.4 kHz based on the beam characterization curves (see fig. 2a,b). This frequency was chosen to be within the broad resonance peak seen in fig. 2a and to be above the linear resonant frequency (frequency of maximum oscillation amplitude at the lowest forcing amplitude) of 492.5 kHz, to ensure the existence of a driving voltage around which the amplitude response (fig. 2b) was sufficiently nonlinear but without hysteresis. Due to its Duffing nonlinearity and the resulting multi-stability of its dynamics, the oscillation amplitude of our MEMS beam can exhibit jumps (up and down) and hysteresis [10]. For the results presented in this paper, we chose a combination of drive amplitude and drive frequency to operate the MEMS beam in a nonlinear regime without hysteresis. The ESN demonstrated in this paper also used an hyperbolic tangent activation function, that was nonlinear and without hysteresis. The beam driving signal frequency was not systematically optimized, as the aforementioned choice gave satisfactory results, but further optimization could lead to better performance. As an example, figure 11 from reference 10 shows that, for a similar system performing a different task, adjusting the drive frequency becomes increasingly important in order to retain good performance as the difficulty of the task is increased. The bounds for the random selection of the other hyperparameters were set by manually exploring the hyperparameter space while looking at the reservoir activations, and keeping only the range of parameters for which the activations were correlated with the walking accelerations.”

The selection of the amplitude of the MEMS forcing signal was also clarified by removing the driving voltage carrier amplitude parameter V_d and instead parameterizing it by expressing the mask values (m_1 , m_2) as applied driving voltage amplitudes (V_1 and V_2) for the two mask levels when no feedback is applied ($\alpha=0$). This way, the results are more readily reproducible since the expression for the driving signal no longer depends on the details of the specific amplitude modulation implementation (which was the reason why m_1 and m_2 had different units for different columns in table SI1). This also better defines the regions of figure 2a,b explored by the MEMS beam during normal operation of the system. Section SI5 was adjusted to take this change into consideration, with the following changes:

(SI page 6, line 17):

“The hyperparameters of the MEMS system that could be adjusted were: the driving voltage frequency (f_d) and amplitude at the two levels of the binary mask (V_1 and V_2) and frequency (f_d), the delayed feedback gain (κ), the leaking rate (α), the binary mask low and high values (m_1 and m_2), the number of virtual nodes (N), the regularization parameter (β) and the virtual node temporal separation (θ).”

(SI page 8, line 16):

“This whole procedure was repeated for 11 random hyperparameter sets which explored different random values of the drive voltage levels V_1 and V_2 , feedback gain κ_7 and leaking rate α and mask values m_1 and m_2 .”

(table SI1, above line 1):

Deleted first row (V_d)

Replaced 5th and 6th rows by:

V_1	77.0 V	78 V
V_2	105.6 V	91 V

(SI page 9, line 9):

“[...] cause a pull-in of the inertial mass due to higher acceleration peaks at these speeds, the values of V_1 , V_2 and κ were slightly lowered [...]”

(SI page 9, line 17):

“[...] most of the hyperparameters were kept identical to those of the reference system (see right column of table SI1), except for slightly different driving voltage amplitudes V_1 and V_2 , and for the feedback gain κ which was set to 0.05. These parameters had to be readjusted in order to accommodate the few differences between the two systems: ± 10 V analog signal rails and 16-bit ADC codes converted to (floating point) voltages for the reference system interfaced with the PC, instead of signals limited to ± 5 V on the real-time system, with unconverted 12-bit ADC integers used as the virtual node activations.”

The optimization of the hyperparameters, especially in a physical system like our MEMS, is still an open research question. We are not aware of a formal methodology that could be efficient enough to be used when tuning the device in real-time (with a person walking on a treadmill). This is why the (admittedly sub-optimal) procedure described in SI5 was used, to obtain functional hyperparameters that were good enough to demonstrate the MEMS device in a real-world task.

Q3: In the manuscript, the page 8 line 28, the performance for the MEMS classifications will reduce if the walking speed is not fixed. In practice, the pace varies greatly. It would be better if the author can provide some suggestions to improve the robustness to wide walking speed range.

Actually, our range of walking speeds is quite larger than what should be expected from a typical population. This was clarified with the following addition to the manuscript (page 9, line 13):

“Relatively slow walking speeds, between 0.36 m/s and 0.72 m/s, were set on the treadmill to correspond to a range of speeds in a population with knee pain. This range ($\pm 33\%$ around median) is much larger than the expected range of natural variations for a given person (e.g. $\pm 3\%$ from reference 34).”

Our on-going research shows that good gait classification performance can be achieved under ecological conditions with a high level of variability (walking speed, but also walking up/down slopes, climbing stairs, walking on hard/soft surfaces, etc.) These results should be published only later this year, so we cannot provide a reference yet. As this is not central to the message of this paper, for the sake of simplicity we propose to not go deeper into this topic.

Q4: Can the devices be switched between nonvolatile and volatile types? If so, the authors are suggested to include these detailed information.

We think that the reviewer might be drawing a parallel with memristors, which can be switched between volatile and non-volatile by applying currents. We are not aware of a similar concept in MEMS. MEMS have a short intrinsic memory provided by the timescale for oscillations to decay in the resonator, that can be increased using a delayed feedback loop and leaky integrator. This is explained on page 6, lines 17-18, which have been modified for the sake of clarity (see Q1 above).

Q5: The short-term memory characteristics have been also used for in-sensor processing other physical stimulation, e.g., Nature Nanotechnology, 2023, 18, 882-888. The authors are suggested to refer to these prior arts.

We thank the reviewer for this interesting, recent paper. It was added as reference 48 (page 17, lines 20-22):

"The moving average and leaky integration increase the short-term memory of the system, which has been shown to be useful for motion classification in other in-sensor systems 48."

Reviewer #2 (Remarks to the Author):

It is a detailed and complete work on the wearable microfabricated accelerometer with the capability of analyzing the human gait. But as a so-called in-sensor device, some problems are supposed to be elaborated in more detail. Although it is not a disruptive innovation considering your previous works, it is an attractive and valuable work if the authors can consider the following comments.

1. In Figure 2b, I don't find the bifurcation points that often exist in nonlinear MEMS resonator. To the best of our knowledge, the bifurcation points and the nonlinear region are vital for MEMS-based reservoir computing.

We found that better performance was achieved by operating the MEMS outside of the hysteresis regime (where jumps in the frequency-response curve can be observed). Even outside the hysteresis regime, there is still a nonlinearity. This nonlinearity is needed to perform nonlinear tasks, but hysteresis is not needed in general. A good example of this would be the ESN, where the neurons are just tanh functions with no hysteresis. An ESN can nevertheless perform various tasks, including the classification of walking patterns demonstrated here.

The manuscript was modified to make this clearer:

(page 18, lines 13-15):

Finally, some of the parameters of the MEMS system had to be tuned to obtain good performance. The tuning procedure for these MEMS hyperparameters is described in the Supplementary Informations, section SI2.

(SI page 8, lines 22-27):

Due to its Duffing nonlinearity and the resulting multi-stability of its dynamics, the oscillation amplitude of our MEMS beam can exhibit jumps (up and down) and hysteresis [10]. For the results presented in this paper, we chose a combination of drive amplitude and drive frequency to operate the MEMS beam in a nonlinear regime without hysteresis. The ESN demonstrated in this paper also used an hyperbolic tangent activation function, that was nonlinear and without hysteresis.

In our MEMS, the clearest indication of a strong nonlinearity is the frequency response “jump” visible in figure 2a (e.g. in the black curve, around 249 kHz). This jump results from the coexistence of two stable solutions in the dynamics of the Duffing oscillator, that can be achieved depending on the direction the drive frequency is swept (increasing or decreasing). Figure 2a and 2b only show the response to increasing beam drive frequency and amplitude sweeps, respectively, as stated in the first paragraph of section SI1. The bistability can be most clearly observed in the following figure (from reference 10):

FIG. 4. Top: response of the beam to a drive frequency sweep for a fixed drive amplitude of 80 V. Bottom: response to a drive amplitude sweep for a fixed drive frequency of 115.1 kHz. All curves show the response at 4 times the drive frequency.

When operating our MEMS, we do not use the jumps, which are a bit too “extreme” to get stable results, but rather the “milder” nonlinearity of the oscillator. We explicitly show in the paper that the gait patterns are not linearly separable (figure 3, logistic regression), and therefore that nonlinearity in the MEMS is required for this task. This point is made on page 10, lines 23-25, including the sentence “This establishes that only nonlinear classifiers can perform well under the conditions of our gait identification task, and further illustrates the usefulness of sophisticated in-sensor processing for this application.”

2.As a so-called in-sensor device, I wonder that how you implement the experiment “in the mechanical domain” with DAC in real-time then? Did you provide a real-time processing in your system? According to your manuscript, I don’t know the detail training data types, I guess it might be mechanical-to-digital that needs a DAC or ADC, so I think it is incapable of accounting for the real-time processing.

We have carefully reviewed our manuscript and made multiple modifications to sharpen our definition of computing “in the mechanical domain”, and to better explain the need for peripheral electronics to solve the very complex gait classification task with the current state of the technology.

We identify below every place in the manuscript where we refer to computing “in the mechanical domain”, and we show the adjustments made to the text.

Page 2, line 8: “By virtue of their operating in the mechanical domain, mechanical computing devices are especially interesting for their integration with sensors that measure sound, acceleration, strain, shape, adsorbed mass or other mechanical properties.”

← This was left as is, as it is a general statement for ‘mechanical computing devices’

Page 2, line 24: “We present the first demonstration of a complete sensor with built-in machine learning capabilities, via a wearable accelerometer that uses neuromorphic computing in the mechanical domain to detect subtle gait patterns in human subjects.”

← This was left as is, but we now define what we mean by “neuromorphic computing in the mechanical domain” in the next sentences (in green, below).

Page 2, line 27: We have clarified how our device operates, with the following new text:

“Our device implements its sensing and nonlinear computing functions using **the displacements of mechanical structures**, and as a result it **can** perform machine learning on sensor data in an especially efficient manner.”

Page 3, line 15: **“As it is commonly done in the field of *physical reservoir computing* 12, 13, the auxiliary electronic components are used to drive our device and to increase the dimensionality of its acceleration data representation using a feedback mechanism.”**

Page 3, line 27: **“All the raw physiological data are processed in the mechanical domain within the device, where they are nonlinearly transformed through the dynamics of the MEMS, before they are digitized. The digital data in the electronics feedback mechanism cannot be inverted to retrieve the raw accelerations because of the complexity of the dynamics of the MEMS, thus providing built-in data security.”**

← We have added the text in green to clarify what we meant and to answer to Reviewer #3, Comment 2 (below).

Page 5, line 3: “We address a human gait classification task (fig. 1) with a microelectromechanical (MEMS) accelerometer that performs both sensing and **reservoir** computing in the mechanical domain.”

← We have removed the word “reservoir”, to clarify that while our ‘reservoir’ is mechanical, some of the reservoir ‘computing’ that we do (e.g. the linear combination of activations) is currently done in electronics.

Page 5, line 13 to page 6, line 1: We have expanded our discussion on the electronics components in the system, to hopefully remove any ambiguity about the structure of our device. The modifications to the text are shown in green below:

“ In order to increase the complexity of the signals that can be generated from the amplitude response of the beam, a feedback technique **that is commonly used with physical reservoir computers** 12 is employed to create multiple different virtual responses using a time-multiplexing technique (fig. 1b), each being a different nonlinear function of the accelerations. The virtual

responses are sampled at regular intervals with conventional electronics, to generate a vector of ‘activation values’ at each time interval (a complete block diagram of the system is shown in the Supplementary Informations, figure SI2). The scalar product between this vector and a trained weights vector is finally computed in a conventional microcontroller at each time interval to produce the output classification for the type of gait (fig. 1b).

The feedback approach to generate virtual activation values has been used successfully in a wide variety of physical reservoir computers⁹, as it allows to adjust the memory of a physical reservoir (which is often fixed by the hardware) to the memory required to solve a task^{17,10}, and as it allows to leverage the fast or energy-efficient physical reservoirs to perform complex nonlinear computations, that are then efficiently processed by conventional electronics. As we demonstrate in this work, even sub-optimized physical reservoir computers that rely on a feedback technique can achieve performance levels that are competitive with conventional electronics (see references 18,19,20 for other examples).”

Page 6, line 8: “For our MEMS device, the in-sensor computing ~~that~~ is done in the mechanical domain, ~~and~~ is the result of the modulation of the high-frequency driving voltage (around 250 kHz) that is the input to the beam reservoir computer, by the low-frequency displacement (below 400 Hz) of the proof mass, which is sensitive to external accelerations.”

← We have added the word “that”, to highlight that not all computing is necessarily done in the mechanical domain.

Page 13, line 9, to page 14, line 1: We have created a separate discussion on the elimination of the feedback. The text now reads:

“More complex design modifications could further reduce the power consumption below 12 mW, for instance by cointegrating the control electronics with the MEMS,³³ ~~or increasing the quality factor of the MEMS resonator with vacuum packaging³⁴ or using the proposal from reference 35 to eliminate the feedback circuitry to create activations entirely in the mechanical domain by using multiple resonators.~~ (see Methods section M6 and Supplementary Informations section SI7 for details).

Lower power consumption levels could arguably be achieved by eliminating the feedback circuitry to create activations entirely in the mechanical domain. This could be achieved using multiple resonators⁴² or multiple proof masses⁴³. A completely mechanical implementation with the number of activations (approximately 100) and memory (on timescales on the order of seconds) required for complex time-series processing (such as gait analysis) has yet to be experimentally demonstrated, but hybrid systems with a few resonators or proof masses^{40,21} could be a stepping stone toward this goal, that proportionally reduces the power consumption of the feedback electronics. Any fully-integrated system should also have efficient drive and read-out mechanisms with efficient electrical implementations⁴¹. Finally, a complete MEMS system should implement a mechanism to adjust the weights in the linear combination of all its activation functions, to support learning. It is unlikely that this could be achieved by tuning the structure of the MEMS, due to the economical costs of creating lithographic masks and,

more fundamentally, to the inherent variability of the fabricated devices. An intriguing possibility would be to perform the linear combination on analog signals instead of in the digital domain, using an array of adaptable elements to adjust the weights, such as memristors⁴². The nonlinear processing of the physical signals could be performed efficiently by MEMS resonators with a high quality factor, while low currents would flow from the memristor array into a summing amplifier with a high input impedance.”

Page 14, line 3: “We have described a device that is both an acceleration sensor and a neuromorphic computer, and that performs its sensing and its nonlinear computing functions in the mechanical domain via the displacement of suspended microstructures.”

← This was left as is, as nonlinear computing is indeed done in the mechanical domain.

Page 14, line 20: “Our demonstration of a single device that solves a complex real-world task by performing both sensing and neuromorphic computing functions directly in the mechanical domain can thus be considered an important milestone for the broad deployment of sensors and machine learning capabilities in emerging applications for wearable medical devices and for the internet of things.”

← This was left as is (the “neuromorphic computing functions” are non-specific).

For the second part of the comment on real-time computing, we do indeed perform gait classification in real-time, by providing feedback as a user walks (please see the beginning of section SI4, and the video in the Supplementary Informations section). There is a small delay between a change in the gait and the corresponding change in the classification, because of the signal averaging process (described in section M5).

3. How many virtual nodes (activations in your manuscript) and how long delay length do you need for different tasks? Or for your providing task, what will they affect your results?

All the results presented in this manuscript were produced with $N=100$ virtual nodes, each with a duration of $\theta=70 \mu\text{s}$, as shown in table SI1. The delay length was set to $N\theta = 7 \text{ ms}$.

These two hyperparameter values were previously found to produce adequate performance in our system for a number of different tasks and were not optimized specifically for the gait analysis task. This is now better explained in the text, in section SI5 (SI page 6, line 20):

“In order to facilitate hyperparameter optimization, the dimensionality of the search space was reduced by using fixed values of $N=100$ virtual nodes and $\beta=1 \times 10^{-6}$ for both the MEMS and the ESN systems, and by using a virtual node separation of $\theta=70 \mu\text{s}$ that yields a delay of $N\theta=7 \text{ ms}$ for the feedback loop in the MEMS system. These parameter values were selected for their general compatibility with the hardware limitations and the timescales of the gait classification task, and from experience with other biomechanical tasks. These other hyperparameters were adjusted with one subject (who was not part of the dataset used in this study) walking on the treadmill at 0.54 m/s, using a bootstrapping approach to estimate the

expected value and standard deviation of the mean ROC AUC for each hyperparameter set with a relatively small amount of data.”

4.1 notice the resonance is around 247kHz in Figure 2, but why did you use “a high-frequency driving force (around 500kHz)”? Please elaborate your tuning process of operation point and the influence of frequency on result in more details.

The electrostatic force on the beam is proportional to the square of the drive voltage, so the time-varying portion of the force driving the beam’s oscillations is actually at twice the drive frequency. As this is not obvious, we have increased the physical relevance of Figure 2 by plotting the frequency responses against the forcing frequency (instead of the drive frequency), so they now peak around 500 kHz. Please see the new figures in the response to Reviewer 1, Q2.

The manuscript was also corrected to read (page 6, line 8):

“For our MEMS device, the in-sensor computing **that** is done in the mechanical domain, **and** is the result of the modulation of the high-frequency driving **forcevoltage** (around **500250** kHz) that is the input to the beam reservoir computer, by the low-frequency displacement (below 400 Hz) of the proof mass, which is sensitive to external accelerations. **As the electrostatic driving force is proportional to the square of the drive voltage, the beam is driven into fast in-plane oscillations (around 500 kHz).**”

Also see the response to comment 2 from reviewer 1, as we have added details of the forcing frequency selection and discussed optimization of the other hyperparameters.

Finally, to improve the reproducibility of our results, we have decided to add critical dimensions for our design to section M1, to complement the SEM images in figure 2e (page 15, line 3):

“The suspended inertial mass and doubly clamped beam were defined by photolithographically patterning their shapes in the AZ MIR 701 photoresist, deep reactive-ion etching (Bosch process) of the 50 μm device layer of a P type (boron dopant, 0.02 $\Omega\text{ cm}$) silicon on insulator (SOI) substrate, and then releasing by HF vapour etching of the 4 μm buried oxide (BOX) **through perforations in the proof mass (10 μm -side squares on a pitch of 20 μm),** while their anchors **(minimum width of 35 μm)** remained attached to the 350 μm thick handle layer due to their larger surface area and absence of perforations. The device was finalized by a metallization step where a stainless steel laser-cut stencil attached over the die allowed the deposition of a Cr-Au film over the electrical traces using electron-beam evaporation. More details of the fabrication process are given in reference 15.

The 50 μm thickness of the device layer provided sufficient out-of-plane stiffness to prevent the pull-in of the inertial mass towards the handle across the 4 μm air gap left by the release step (etching of the BOX), but more importantly provided substantial mass **(49 μg)** in a small footprint **(1 mm^2),** thus providing a sufficient sensitivity to uniaxial accelerations. Four pairs of **thin 3 μm -wide** folded flexure springs **(436 μm and 488 μm long segments)** were used to suspend the inertial mass. They were tuned to provide sufficient longitudinal displacement in

reaction to accelerations (6.8 N/m spring constant), while restricting transverse and rotational displacements. A 8 μm -wide electrostatic transduction gap separated the inertial mass from the beam oscillator. The beam geometry was width (4 μm) and length (300 μm) were adjusted^{15, 13} for fast oscillations (much above the resonance frequency of the inertial mass, in a range where the resonator is not sensitive to inertial forces), as well as to easily achieve nonlinear dynamics through mechanical stiffening at large displacements. Thin 1.5 μm by 12 μm piezoresistive strain gauge pairs attached to the beam were included in the design to allow the differential piezoresistive measurement of its oscillations. Sets of stopper structures (18 μm diameter half-discs) surrounded the inertial mass in order to limit its range of motion to 5 μm , to protect the device from shocks and electrostatic pull-in which could otherwise damage the device by short-circuiting the capacitive transduction gap.”

Reviewer #3 (Remarks to the Author):

This paper presents an incremental advancement of the author's previous work for using MEMS RC computing. Specifically, besides using this RC for gait classification, most of the technical work and demonstration and hardware were already demonstrated in the author's previous paper in the 2020 JMEMS journal. Below are the concerns the authors need to address:

1-The sentence "using displacement degrees of freedom that are in the same physical domain" in page 2 is not clear.

Please refer to comment 2 from Reviewer #2 for the modification to the manuscript.

2-All raw physical data are processed in the mechanical domain, is not completely true. My understanding of the virtual neuron approach is that those samples need to be stored and processed with a microcontroller to provide the classification. Along this line how do you justify still using microcontrollers with this MEMS computing approach?

Please refer to comment 2 from Reviewer #2 for the detailed answer to the second part of your question.

For the first part, the text was clarified in the following way (page 3, line 26):

"ImportantlyIn addition, only the classification labels are transmitted (wirelessly) by our in-sensor computing device. All the raw physiological data are processed in the mechanical domain within the device, where they are nonlinearly transformed through the dynamics of the MEMS, before they are digitized. The digital data in the electronics feedback mechanism cannot be inverted to retrieve the raw accelerations because of the complexity of the dynamics of the MEMS, thus providing built-in data security. This feature of in-sensor computing could be especially relevant for wearable medical devices, to protect the privacy of their users and to save energy by reducing the amount of transmitted data and the use of encryption algorithms."

3-I am not sure the energy efficiency is a valid claim given as demonstrated in the paper a microcontroller consumes less power to perform similar classification without the RC approach.

Our argument on energy efficiency is presented in the third paragraph of the section 'In-sensor computing for wearable devices' (page 12, line 22, to page 13, line 13). We were able to build a functional prototype of a MEMS wearable device that solves a rather complex, nonlinear classification task in real time. The raw power consumption of this prototype was 970 mW. A detailed analysis presented in sections M6 and SI7 shows that the power consumption could be reduced to 94 mW through straightforward optimization. We believe that this figure of 94 mW is the most appropriate comparison for the 280 mW used by a conventional microcontroller solution running a software ESN, as said microcontroller solution is very heavily optimized, as it

is the result of literally multiple decades and billions of dollars of R&D. As such, we believe that it is fair to say that our MEMS in-sensor approach *is* energy efficient. Even lower power levels should be achieved for our complex gait analysis task with more substantial improvements (as reported, around 10 mW).

We were careful to rigorously report all of our measurements, hypotheses and calculations in sections M6 and SI7, so that the correctness of our energy efficiency claims can be checked and validated independently.

4-In page 2, line 6 it refers to the sensitivity curve in Fig.2c, and nothing there is about that.

This was a referencing error which has been corrected. It now reads **fig.2d** (SI page 2, line 14).

5-Not sure what the point of claiming 99% accuracy on page 9 is, if the individual classifier was trained for each user. To me, this sounds like a standard overfitting problem. Typically, you need to build a general model for all the users and then test it over them.

One modality that we are exploring with our physiotherapist colleagues is to use the MEMS device for gait retraining: a MEMS device is trained for an individual with a functional limitation walking on a treadmill, and is later used to provide biofeedback in daily life to alert the user of an improper gait pattern. In this case, it is clinically relevant to achieve high accuracy on a single individual.

Most importantly, we wanted to show the evolution of the performance of the MEMS while going from a simpler (albeit already quite complex) task with a single individual, to a much more difficult task (multiple speeds, multiple individuals) – see figure 3.

This was clarified in the text in the following manner (page 10, lines 4-9):

“Our results show that the TO and the TL gait detectors could both perform very well, with a [...] TPR above 99% and a typical probability for **in**correctly **not** identifying the TO pattern while the subject was not walking with that pattern (FPR) below 1% (respectively 90% and 10% for the harder TL pattern, see fig. 1c). **This level of performance for a training customized to a single subject could be useful in the context of personalized medicine** ³⁵. Figure 3a **further** shows the AUC for subtasks with various levels of complexity.”

Considering overfitting, we draw the attention of the reviewer to the last paragraph of section M4, where the split between training and testing datasets is described. We made sure to avoid any overfitting by evaluating the performance on a validation set (not used in training).

6-How do we justify the high required voltage for operation?

The high voltage driving signal is required to drive the MEMS beam of the current design into a sufficiently nonlinear regime (see Fig.2a,b). In future designs the beam geometry could be adjusted to eliminate this requirement.

This was clarified in the text with the following modifications (SI page 12, lines 6-17):

“After these design modifications, the most important remaining sources of power dissipation will be the high-voltage amplifier with over 50% of the total dissipation, followed by the voltage regulator, the microcontroller and the wheatstone bridge polarization. More complex design modifications could further reduce power consumption. ~~Examples~~ The high voltage driving signal is required to drive the MEMS beam of the current design into a sufficiently nonlinear regime. Adjustments to the beam geometry (such as reducing its width from 4 μm to 1 μm and thickness from 50 μm to 15 μm) could reduce the driving voltage to below 5 V and eliminate the high-voltage amplifier. Other examples of possible design modifications include changing the beam displacement measurement mechanism from piezoresistivity to capacitive transduction, ~~beam geometry adjustments to reduce the driving voltage and eliminate the high-voltage amplifier,~~ vacuum packaging to increase the quality factor of the mechanical structures, and getting rid of the microcontroller by implementing the delayed feedback loop using a charge-coupled device or by coupling the degrees of freedom of multiple physical oscillators.”

7-Authors need to acknowledge a large body of research on using MEMS for computing, one recent of which was published recently in the COMMSSENG journal where a MEMS computing approach was presented without the need for a microcontroller and thus achieving near zero power (Energy efficient integrated MEMS neural network for simultaneous sensing and computing).

Thank you for the interesting reference. It's indeed highly relevant and was added to our manuscript as reference 43 (page 13, line 16; see Reviewer #2, Comment 2).

Reviewers' comments:

Reviewer #1 (Remarks to the Author):

The authors have responded the questions in a satisfactory manner. I would recommend the acceptance in present form.

Reviewer #2 (Remarks to the Author):

It is appreciated that the authors have elaborated and answered every specific comment or question of the reviewers. Only one question but not essential that I still want to discuss with the authors, which are listed below.

1.Thanks to the authors for elaborating and answering my previous comment 2, and for a detailed discussion and explanation for the issue on “in the mechanical domain” in their revised manuscript. However, as a delayed feedback physical reservoir computing system, I think discretizing and digitizing as well as mask operation for the raw data are parts of pre-processed. So, I don't share the opinion with authors on “All the raw physiological data are processed in the mechanical domain within the device” in page 3, line 27. But don't take this comment seriously, I know it is hard for a delayed feedback RC because the mask operation for the raw data and delayed loop are necessary. The authors have also demonstrated that “Lower power consumption levels could arguably be achieved by eliminating the feedback circuitry to create activations entirely in the mechanical domain.” So, my previous question about the “training data type” just intend to ask if there are other ways to preprocess the raw data “in the mechanical domain”. Now, I am clear.

2.Besides, I think authors' revisions to the reviewer's comments are too detailed to far from an “article” publication. Of course, I think these revisions are valuable, some can be arranged in the supplementary and some can be compacted and reformulated. The authors are supposed to rearrange and polish their manuscript to present a better layout for publication.

Reviewer #3 (Remarks to the Author):

I think the authors had made significant changes to improve the paper. I still have the following minor comments:

again, I'm not completely sold out on the power comparison between the reported work and using a microcontroller. An alternative is to focus on or highlight other aspects of this, like it may eliminate the need for the sensor conditioning circuit to read acceleration or any other advantages of using MEMS as an RC. Also, with respect to the personalized modeling approach, do we except that we need to train this RC for each user? dos this impact the usability of this hardware?

We thank the editor and reviewers for the additional comments and suggestions to improve our manuscript. Their time and expertise is greatly appreciated.

The reviewers' comments are copied verbatim below (italicized). Every specific comment or question was addressed with the modifications that are described below, and that are identified in the revised manuscript in green. Our references to page and line numbers are for the revised manuscript.

While reviewing the text we realized that many references to the sub-panels of figure 2 had not been properly updated after the first revision of the manuscript. This was corrected in the latest revision.

Reviewer #1 (Remarks to the Author):

The authors have responded the questions in a satisfactory manner. I would recommend the acceptance in present form.

We thank the Reviewer for his or her time.

Reviewer #2 (Remarks to the Author):

It is appreciated that the authors have elaborated and answered every specific comment or question of the reviewers. Only one question but not essential that I still want to discuss with the authors, which are listed below.

1.Thanks to the authors for elaborating and answering my previous comment 2, and for a detailed discussion and explanation for the issue on “in the mechanical domain” in their revised manuscript. However, as a delayed feedback physical reservoir computing system, I think discretizing and digitizing as well as mask operation for the raw data are parts of pre-processed. So, I don't share the opinion with authors on “All the raw physiological data are processed in the mechanical domain within the device” in page 3, line 27. But don't take this comment seriously, I know it is hard for a delayed feedback RC because the mask operation for the raw data and delayed loop are necessary.

We have further clarified the text to avoid any misunderstanding related to the definition or wording that we have initially used. The text now reads (lines 1-2, page 4):

“In addition, only the classification labels are transmitted (wirelessly) by our in-sensor computing device. **All the raw physiological data are processed in the mechanical domain within the device, where they**The raw physiological data (*i.e.*, the accelerations) are nonlinearly transformed through the dynamics of the MEMS, before they are digitized. The digital data in the electronics feedback mechanism cannot be inverted to retrieve the raw accelerations because of the complexity of the dynamics of the MEMS, thus providing built-in data security.”

This reformulation is more concise and it carries the same meaning as the original text, albeit more clearly.

The authors have also demonstrated that “Lower power consumption levels could arguably be achieved by eliminating the feedback circuitry to create activations entirely in the mechanical domain.” So, my previous question about the “training data type” just intend to ask if there are other ways to preprocess the raw data “in the mechanical domain”. Now, I am clear.

We thank the Reviewer for this confirmation that our answer to the comment from the previous review was satisfactory.

2. Besides, I think authors’ revisions to the reviewer’s comments are too detailed to far from an “article” publication. Of course, I think these revisions are valuable, some can be arranged in the supplementary and some can be compacted and reformulated. The authors are supposed to rearrange and polish their manuscript to present a better layout for publication.

We have reviewed the main text, and decided to move to Methods section M6 the last paragraph from the section “In-sensor computing for wearable devices”. This paragraph was added at the last revision in response to the comments from the Reviewers. While this does not change in any way the content/message of the paper, we think it creates a better flow for the main ideas.

We have added (page 13, line 12):

“The lowest power consumption could arguably be achieved by eliminating the feedback circuitry to create activations entirely in the mechanical domain, and by forming the linear combination of the activations in the analog domain, as discussed in Methods section M6.”

The last paragraph of the section “In-sensor computing for wearable devices” was moved to page 22.

Reviewer #3 (Remarks to the Author):

I think the authors had made significant changes to improve the paper. I still have the following minor comments: again, I’m not completely sold out on the power comparison between the reported work and using a microcontroller. An alternative is to focus on or highlight other aspects of this, like it may eliminate the need for the sensor conditioning circuit to read acceleration or any other advantages of using MEMS as an RC.

We have clarified the message by providing a better explanation of our assumptions and methodology for the power comparison earlier in the paper. The Introduction now reads (lines 16-27, page 3):

“Our prototype device is shown to consume power at a level that is similar to a heavily-optimized commercial microcontroller chip implementing the same gait classification task in software (section “In-sensor computing for wearable devices”). As shown below, The total power consumed by the MEMS prototype device was measured and compared to a model summing the calculated power consumed by each subsystem (970 ± 10 mW total measured power, 958 mW calculated, see Methods section M6). This analysis showed that a simple re-design of

certain sub-systems in our prototype device would further reduce its power consumption by an order of magnitude (down to an estimated 94 mW, see Methods section M6), thus providing a significant advantage over conventional microcontroller-based solutions and paving the way for wearable devices that are much smaller or function much longer on a battery charge (see section “In-sensor computing for wearable devices”). Our results thus firmly establish in-sensor computing in MEMS accelerometers as a competitive alternative to conventional embedded software solutions, in a device that is integrated and effective enough for a gait classification task that is difficult and relevant for real-world applications.”

This reformulation carries the same meaning as the original text, but it draws the attention of the reader to the detailed measurements and calculations reported in the Methods section earlier in the paper, to reinforce our conclusions from the power consumption analysis.

Also, with respect to the personalized modeling approach, do we except that we need to train this RC for each user? dos this impact the usability of this hardware?

The manuscript states that the “training [is] customized to a single subject [...] in the context of personalized medicine”. The manuscript was clarified to highlight that this does not impact usability (lines 9-10, page 10):

“This level of performance for a training customized to a single subject could be useful in the context of personalized medicine 35 , where a device could be trained in a clinical environment to provide a high-accuracy gait classifier to achieve a specific therapeutic objective.”

This idea is explained rather well in reference 35 that was added at the last revision of our paper: “Consider the clinical case of a physiotherapist who detects both an excessive loading rate and increased hip adduction in a runner who develops anterior knee pain. These variables could be adjusted with feedback in the lab to determine the running pattern that keeps the athlete pain free; the runner can be monitored carefully and given feedback by means of inertial measurement units and an integrated ‘smart’ watch so the runner can apply the intervention in the natural environment. This is ‘personalised medicine!’”¹

Our device provides an innovative means to implement precise and efficient gait monitoring that is tailored to a given individual, for instance by training ‘in the lab’. This is precisely the modality described in reference 35 as an ideal, ‘personalized medicine’ approach. For the sake of brevity and because our paper’s focus is on the technology (and not so much the physiotherapy applications), we would prefer not to expand more on this secondary topic.

¹ Napier, Christopher, Jean-Francois Esculier, and Michael A. Hunt. "Gait retraining: out of the lab and onto the streets with the benefit of wearables." *British journal of sports medicine* 51.23 (2017): 1642-1643.